# The Association of Life Events Outside the Workplace and Burnout: A Cross-Sectional Study on Nursing Assistants

**DOI:** 10.3390/ijerph19159342

**Published:** 2022-07-30

**Authors:** Mariana Tortorelli, Telma Ramos Trigo, Renata Bolibio, Camila Colás Sabino de Freitas, Floracy Gomes Ribeiro, Mara Cristina Souza de Lucia, Dan V. Iosifescu, Renério Fráguas

**Affiliations:** 1Grupo de Interconsulta, Departamento e Instituto de Psiquiatria, Hospital das Clínicas, Faculdade de Medicina, Universidade de São Paulo, São Paulo 05403-903, Brazil; mfptortorelli@gmail.com (M.T.); trtrigo@yahoo.com.br (T.R.T.); camilacolas@gmail.com (C.C.S.d.F.); 2Divisão de Psicologia, Instituto Central Hospital das Clínicas, Faculdade de Medicina, Universidade de São Paulo, São Paulo 05508-000, Brazil; rebolidio@gmail.com (R.B.); mara.cristina@hc.fm.usp.br (M.C.S.d.L.); 3Gabinete do Secretário, Secretaria de Estado da Saúde de São Paulo, São Paulo 05403-000, Brazil; fgribeiro@saude.sp.gov.br; 4Clinical Research Division, The Nathan S. Kline Institute for Psychiatric Research Orangeburg, New York, NY 10962, USA; dan.iosifescu@nyumc.org; 5Psychiatry Department, New York University School of Medicine, New York, NY 10016, USA; 6Laboratório de Neuroimagem (LIM-21), Departamento e Instituto de Psiquiatria, Faculdade de Medicina, Universidade de São Paulo, São Paulo 05403-903, Brazil; 7Divisão de Psiquiatria e Psicologia do Hospital Universitário, Universidade de São Paulo, São Paulo 05403-903, Brazil

**Keywords:** burnout, Maslach Burnout Inventory, life events, stress, depression, nursing assistants

## Abstract

Background: Burnout, by definition, is related to adverse chronic workplace stressors. Life events outside the workplace have been associated with an increased risk of psychiatric morbidity. However, it is unknown whether life events outside the workplace increase the severity of burnout. Purpose: The aim of the study was to investigate the association between burnout and life events outside the workplace in nursing assistants. Methods: In an observational, cross-sectional, single-site study of 521 nursing assistants at a university hospital, we assessed burnout with the Maslach Burnout Inventory-Human Services Survey, and life events with the Social Readjustment Rating Scale. We constructed equations of multiple linear regression analyses that included each burnout subscale as the dependent variable and a domain of life events as the independent variable. Results were adjusted for potential confounders, including gender, no religion or faith, years of work, and depression. Results: An increase in the number of life events in the domain of personal changes or difficulties (e.g., personal injury or illness, sexual difficulties, change in recreation, church activities, social activities, sleeping habits, eating habits and revision of personal habits) was associated with increased severity of emotional exhaustion. An increase in the number of life events in the domain of changes in familial situation and in the domains of death of relatives or friends were associated with increased severity of depersonalization. Those associations were independent of work-related life events and other potential confounders. Conclusions: Life events outside the workplace may increase the levels of burnout in nursing assistants.

## 1. Introduction

Burnout is a syndrome resulting from prolonged levels of stress at work characterized by emotional exhaustion, depersonalization and decreased personal accomplishment [1,2]. According to the International Code of Diseases-11 edition, (ICD 11) released in 2022, burnout has been coded among the “Problems associated with employment or unemployment” (QD85) [3]. Burnout is destructive to the individual; it has been associated with the intention of giving up nursing [4], social dysfunction, somatic symptoms and symptoms of depression and anxiety [5].

There is no specific study on nursing assistants in Brazil, but registered nurses (RNs) face problems such as lack of recognition of the scientific aspect of nursing, the relevance of nursing work not being adequately recognized by the media, and work overload [6]. Nursing assistants in Brazil, are involved in basic bedside care, as well as more complex activities including inserting bladder and gastric catheters and collecting material for laboratory tests under the supervision of a RN [7,8]. Nursing assistants are potentially more vulnerable to burnout. For example, they do not require a college degree in order to undertake their duties [9], which could preclude the development of skills such as reasoning ability, and a consequent failure in rescuing some patients [10]. The activities of nursing assistants in Brazil appear to be similar to the responsibilities of licensed practical nurses in other countries [11]. Many studies use the category of licensed practical nurse, as it is used in United States of America (USA) (18–24 months of training). However, the category and possibly some duties of such professionals vary across the countries: licensed vocational nurses (one year of training, USA) [12,13]; enrolled nurses (Australia and New Zealand, 18 months of training) [14,15]; and division 2 nurses (18 months of training, Australia) [16]. In 2010, there were 533,422 (36.8%) nursing assistants and 87,119 (19.8%) RNs in Brazil [17]. By April 2022, there were 442,345 (16.73%) nursing assistants and 653,636 (24.72%) RNs [18]. Prior studies have generally focused on heterogeneous samples combining licensed practical nurses and RNs [19]. Considering the peculiarities of nursing assistants, studies should focus specifically on these vulnerable professionals.

By definition, burnout is a work-related condition, and is associated with factors such as increased occupational stress [20], increased workload and decreased nurse-to-patient ratio [21]. In a more inclusive view, burnout may result from the interaction of various factors including personal traits, emotion, work-related stress and previous psychiatric disorders [22,23,24] Consequently it is theoretically possible that episodes of stress from other life domains could contribute to the development of burnout [25]. In fact, some studies have investigated the possible influence of personal stressors on burnout. An association of personal life stressors with burnout has been reported in a sample of Swedish twins [26], in medical residents [27] and in medical students [28]. Among nurses, burnout has been associated with private life problems in one study [29] and with chronic and unresolvable stress not limited to the workplace in another one [25]. Life events not specifically in the workplace have been associated with an increased risk of a psychiatric diagnosis [30]. However, no study has investigated the potential contribution of external life events to the severity of burnout specifically in nursing assistants. Considering the vulnerability of nursing assistants, we expect that life events outside the workplace might significantly increase the levels of burnout in these professionals. If confirmed, knowing that life events outside the workplace increase levels of burnout in nursing assistants might allow the development of programs specifically devoted to assist these workers in dealing with life events and reducing the levels of burnout. This study aimed to investigate the possible influence of life events outside the workplace on levels of burnout in a nursing assistant sample. Additionally, we investigated the presence of potential confounders in that relationship.

## 2. Methods

### 2.1. Study Characteristics

In this observational, cross-sectional, single-site study, we interviewed face-to-face 521 nursing assistants using the Primary Care Evaluation of Mental Disorders (PRIME-MD). We assessed burnout with the Maslach Burnout Inventory-Human Services Survey (MBI-HSS) and life events with the Social Readjustment Rating Scale (SRRS).

### 2.2. Participants and Procedures

Data from this sample regarding the influence of depression on validation of the MBI-HSS, but not about life events, have previously been reported elsewhere and in a master thesis [31,32]. In short, nursing assistants were assessed face-to-face by a psychiatrist or a psychologist researcher during the period June to November 2009 at the general medical units (i.e., general internal medicine, hematology, endocrinology, pulmonary, nephrology, immunology, rheumatology, gastroenterology, and geriatric units), surgical unit, emergency room and intensive care unit. The researchers went to every unit and asked for volunteers. Equivalent numbers of nursing assistants were recruited from the day shift (i.e., 7:00 a.m. to 7:00 p.m.) and the night shift (i.e., 7:00 p.m. to 7:00 a.m.). Nursing assistants represent 76.9% of the nursing workforce staff, and our sample comprised 35.3% (521\1477) of all nursing assistants when the study was conducted at a General Unit of a University Hospital in Sao Paulo, Brazil [33]. 

### 2.3. Measures

#### 2.3.1. Life Events

Life events are external situations that require psychological readjustment [34]. To assess life events, we used the SRRS, an instrument listing 43 significant life events which may have occurred over the previous twelve months. We categorized life events by their context, as previously performed, attributing a score of “1” for the presence of each event. We categorized the events as work-related (dismissal from work, retirement, business readjustment, change in financial state, change in responsibilities at work, trouble with boss, change in working hours or conditions, change to a different line of work and vacation) and outside workplace: (a) death of relatives or friends (e.g., death of a spouse, death of a close family member and death of a close friend); (b) changes in familial situation (e.g., divorce, marital separation, marriage, pregnancy, gain of new family member, change in number of arguments with spouse, son or daughter leaving home, spouse begins or stops work, change in living conditions, change in residence, change in number of family get-togethers, change in health of family member); (c) changes in environment (e.g., change of school, change of house, change in the number of persons living in the house; (d) personal changes or difficulties (e.g., personal injury or illness and/or sexual difficulties, change in recreation, church activities, social activities, sleeping habits, eating habits and revision of personal habits); (e) legal problems (e.g., imprisonment, major mortgage, foreclosure of mortgage or loan, trouble with in-laws, minor mortgage or loan and minor violation of law). To perform this categorization, from the 43 original items, we did not include some events such as those with evident positive valence (e.g., marital reconciliation and outstanding personal achievement), and those outside the focus of our study (e.g., Christmas).

#### 2.3.2. Burnout

We used the MBI-HSS, Portuguese Version, to assess burnout; it is a self-report instrument consisting of 22 items, grouped in 3 subscales. The scores on each subscale are assessed separately: a 9-item subscale for emotional exhaustion, a 5-item subscale for depersonalization and an 8-item subscale for personal accomplishment. The severity of each item is evaluated on a 7-point Likert scale based on its frequency, ranging from never (score 0) to every day (score 6). Scores for emotional exhaustion, depersonalization, and personal accomplishment ranges are 0–54, 0–30 and 0–48, respectively. Cut-off points have been established based on the U.S.A. normative data or by dividing the subscales into tertiles or quartiles [35].

#### 2.3.3. Major Depressive Disorder

The diagnosis of major depressive disorder (MDD) was performed by a psychiatrist (T.M.T.) using the PRIME-MD [36], Brazilian Portuguese version [37]. It is a five-module structured interview. The mood module presents 9 yes/no questions investigating the depressive symptoms included in the *Diagnostic and statistical manual of mental disorders* (*DSM*) third version revised (*DSM-III-R*) criteria to diagnose major depression. These symptoms have been maintained in the fourth and fifth versions of the *DSM* (*DSM-IV* and *DSM-V*) [38,39].

#### 2.3.4. Socio-Demographic and Occupational Data

Age, gender, race (based on the Brazilian census), marital status, number of children, religion, hours of work in the week, absenteeism, and years of work in the hospital were obtained by a self-report questionnaire.

### 2.4. Ethical Considerations

All subjects participated voluntarily and began the study only after providing a signed informed consent form. An interview could raise awareness of previously dormant issues, which could be a point of concern. Individuals who were sensitized by the interview were oriented regarding treatment options in the institution. The Hospital Committee of Ethics in Research approved the study as involving no or minimal risk.

### 2.5. Statistical Design and Analysis

We report absolute frequency and percentages for categorical variables (i.e., gender, race, marital status), and mean and standard deviation for continuous variables. To investigate the relationship between life events and burnout, we first processed an analysis of total scores of life events with each domain of burnout (i.e., depersonalization, emotional exhaustion and personal accomplishment). For the burnout domains showing an association with total scores of life events, we investigated their association with specific domains of life events (i.e., death of relatives or friends; changes in familial situation; personal changes and difficulties; changes in the environment; work-related events; and legal troubles). We used simple linear regression analyses to investigate the associations without adjustments. Then, we constructed an equation of multiple linear regression to investigate the possible influence of confounders [40]. An equation was constructed for each domain of burnout significantly related to life events as the dependent variable. As the independent variable for each equation, we included the domains of life events that showed an association in the non-adjusted analysis. The equations were adjusted for work-related life events, gender, age, and variables potentially related to burnout showing significant association in the non-adjusted analysis. We used linear regression to investigate the relationship between life events and burnout as it has been used to determine the impact of stress on physical health and work-family conflict [41].

We also performed an analysis dividing each subscale of burnout into tertiles as previously performed [42]. This strategy allows an analysis for those with high levels of burnout (upper tertile) complementing the analysis of the scores without using cut-offs. The cut-offs for lower, moderate, and upper tertiles were, respectively: (a) emotional exhaustion: 0–15, 16–26 and 27–57; (b) personal accomplishment: 41–48, 35–40 and 0–34; (c) depersonalization: 0–2, 3–7 and 8–27. We compared the upper to the lower tertile. Then we constructed equations of multiple logistic regression (with the same rationale described above for multiple linear analyses) and used the domains of burnout analyzed in tertiles as the dependent variable. We used a significance level of <0.05, two-tailed and the statistical software for data science (Stata 13) to perform the analysis. 

## 3. Results

In this observational, cross-sectional, single-site study, we examined the relationship between life events outside the workplace and burnout. We interviewed face-to-face 521 nursing assistants using the PRIME-MD and assessed burnout with the MBI-SS and life events with the SRRS. Of the 521 nursing assistants, most were female (91.2%; female/male = 10.36) and the mean age was 39.4 (±9.7) years; 320 (61.4%) worked in the general internal medicine unit; 60 (11.5%) in the surgical unit; 66 (12.7%) in the emergency room; and 75 (14.4%) in the intensive care unit. We included 345 nursing assistants from the day shift (i.e., 7 a.m. to 7 p.m.), and 176 from the night shift (i.e., 7 p.m. to 7 a.m.). Most nursing assistants were female (91.2%), white (51%), married (50%) and with children (68.5%). The mean age was 39.4 (±9.7) years and the mean time of work at the institution was 8.5 (±6.4) years. The sociodemographic characteristics are presented in Table 1. 

Total score of life events was associated with depersonalization and emotional exhaustion (Table 2), but not with personal accomplishment (Coef. = −0.1055076; Std. Err. = 0.0702751; t = −1.50; *p* = 0.134; 95% Conf. Interval = −0.2435662 to 0.0325511). Thus, bivariate analysis investigating the association with specific domains of life events were performed for emotional exhaustion and depersonalization (Table 2).

The multiple linear regression analysis indicated that burnout was influenced by life events in the domain of personal changes and difficulties, changes in familial situation and death of relative or friends, independently of work-related life events and potential confounders. An increase in the number of life events in the domain of personal changes and difficulties increased the severity of emotional exhaustion (Table 3); an increase in the number of life events in the domain of changes in familial situation (Table 4) and death of relative or friends (Table 5) increased the severity of depersonalization. The multiple logistic regression analysis (results presented in the supplementary material) indicated that for each life event in the domain of personal changes and difficulties, there was an increase of 14% in the chance of being in the upper tertile of emotional exhaustion; for each life event in the domain of death of relatives or friends, there was an increase of 40% in the chance of being in the upper tertile of depersonalization. For each life event in the domain of changes in familial situation, there was an increase of 17% in the chance of being in the upper tertile of depersonalization. The logistic regression analysis models were adjusted for work-related life events and other potential confounders.

## 4. Discussion

In this study with 521 nursing assistants, an increased number of life events outside the workplace was associated with increased severity of depersonalization and emotional exhaustion. The multiple linear analysis showed that an increased number of life events in the domain of personal changes or difficulties was associated with increased severity of emotional exhaustion. An increased number of life events in the domain of familial situation and in the domain of death of relative or friends was associated with increased severity of depersonalization. The adjusted multiple linear analysis showed that these associations were independent of life events in the workplace, age, gender and other potential confounders. 

### 4.1. Life Events in the Domain of Changes in Family Situation and Depersonalization 

Our findings indicating an association between an increase in the number of events in the domain of changes in family situation and an increase in the severity of depersonalization are in line with the theoretical concept of work-family balance. Burnout has been reported to be negatively influenced by family-work role conflicts [43]. Work-family imbalance has been reported to be related to withdrawal intentions among public accountants [44]. Severe work-private life conflicts have been related to high quantitative demands among health professionals leading them to hide their emotions [45]. It should be noted that while burnout was found to mediate the association of family work conflict with workplace injuries, the relationship of family-work enrichment with workplace injuries was mediated by work engagement [46]. For some samples, besides non-work conflict, leisure activities and support in personal life have been identified as predictors of burnout [47]. Considering specifically life events, Mather et al., have already found an association of burnout with stressful life events, including serious family problems, physical illness, and divorce or separation—independently of confounders— among 25,378 Swedish twins from the general population [26]. The association of burnout with personal life events has also been reported in medical students [28], medical residents [27] and university professors [48]. Our findings confirmed the association of burnout with life events outside the workplace in a vulnerable population of nursing assistants. In our sample, for each life event in the domain of changes in familial situation, there was an increase of 17% in the chance of being in the upper tertile of depersonalization (supplementary material). The potential influence of life events outside the workplace on burnout has brought doubt to the concept that burnout is primarily linked only to work-situated factors [47].

Our results reveal the possibility that the association might differ according to the domain of burnout (e.g., emotional exhaustion or depersonalization) and according to the domain of the life events outside the workplace (i.e., life events in the domains of personal changes or difficulties, of changes in familial situation or in the domain of death among relatives or friends).

### 4.2. Life Events in the Domains of Death of Relatives or Friends and Depersonalization 

We found that the domain of events related to death of relatives or friends (i.e., death of a spouse, death of a close family member and death of a close friend) was associated with high levels of depersonalization. The experience of grief may possibly contribute to feeling “numb”, contributing to the development of depersonalization. While there is an absence of studies on nursing assistants, our findings are consistent with data regarding RNs. Depersonalization has recently been associated with fear of death and death avoidance in RNs [49]. Intriguingly, among RNs, coping with death of patients has been associated with a decrease in depersonalization, possibly by an increase in sensitivity [50]. The length of time working with dying patients and personal characteristics may contribute to the direction of the relationship. 

### 4.3. Life Events in the Domain of Personal Changes or Difficulties and Emotional Exhaustion 

We found an association of an increase in the number of life events in the domain of personal changes or difficulties (e.g., the domain including personal injury or illness, sexual difficulties, change in recreation, church activities, social activities, sleeping habits, eating habits and revision of personal habits) and increase in the severity of emotional exhaustion. These results suggest that life events in church activities increase the severity of depersonalization. We also found that having a religion decrease the severity of depersonalization. The relevance of church activities is supported by previous findings showing that religious faith has a protective role for the development of burnout [51]. Confirming such relevance, in our sample, no religion of faith was also significantly associated with high levels of emotional exhaustion. Considering changes in sleeping habits, included in the domain of personal changes or difficulties, for the general population, in the same line of our findings, insomnia has been reported to increase the risk for persistence of emotional exhaustion [52]. In RNs, burnout has previously been related to sleep problems such as short periods of sleep [53]. 

The cross-sectional design of our study precludes any determination of causal direction between the events in personal changes or difficulties outside the workplace and increased levels of emotional exhaustion. Longitudinal studies support transactional models of stress, indicating that stressors and stress mutually influence each other [54]. Therefore, both directions are possible and relevant.

### 4.4. Depression, Burnout and Life Events

We found that the association of emotional exhaustion with changes in the family context and with personal changes or difficulties was explained by MDD (and other confounders). Major depressive disorder was associated with all burnout subscales, consistent with previous studies [55]. Consequently, our findings support the view that symptoms of burnout should be considered a trigger to screen for MDD [56]. Actually, burnout and depression have been shown to largely overlap in their symptoms, making it difficult to clinically establish a difference between the two syndromes [57]. Our findings, showing that high levels of emotional exhaustion were associated with life events independently of MDD, support the relevance of the concept of burnout as an independent entity. However, data showing the significant overlap between exhaustion and MDD have endorsed the view that burnout could be considered a form of depression and not a distinct disorder [57]. In this line, the association of burnout with non-work life events, as we found here, supports the view that the concept of burnout, exclusively defined as a work-related condition, has some weaknesses [58]. Supporting the relevance of factors related to the person instead of only those related to the work condition, Bianchi et al., found that burnout was not more strongly linked to organizational and work-contextualized variables than to personality traits [59].

### 4.5. Burnout, Life Events and the COVID-19 Crisis

It should be considered that changes in contemporary external conditions including the COVID-19 pandemic crisis could interfere with the occurrence of life events and burnout. During the COVID-19 crisis, high burnout level was reported for depersonalization, emotional exhaustion, and personal accomplishment, respectively, in 47%, 46%, and 48% of 1925 emergency medicine professionals including physicians, nurses, and paramedics [60]. In Brazil, a great impact of COVID-19 in depersonalization and emotional exhaustion has been reported in a sample composed of physicians, nurses, nurse technologists and physiotherapists [61]. During the COVID-19 pandemic, it has been found that the highest average scores and counts of stressful life events and highest work-family conflict levels were reported respectively, by nurse practitioners and registered nurses compared to physicians and other professionals [41]. In addition, students in the nursing field experienced more traumatic events during the COVID-19 pandemic crisis than before it [62]. The sense of coherence may act as a negative regulator between burnout and depression and it has been reported to be reduced by the influence of the COVID-19 pandemic [63]. Our study was developed before the COVID-19 pandemic crisis; thus, it is possible that during the COVID-19 crisis and also the post COVID-19 crisis, other specific life events could show a relationship with burnout.

## 5. Limitations

Some limitations of our study should be considered. Our sample comprises nursing assistants from a single site, a teaching hospital. Studies are needed to confirm the extension of our findings for nursing assistants in other settings. We asked for volunteers from each unit until completing the relative proportion of nursing assistants from that unit. Thus, we are not able to estimate the percentage of nursing assistants that may have been interested in participating after we completed our target of responders from each unit. The cross-sectional design of our study limits causal inference of the relationships reported, and longitudinal studies are necessary to investigate whether increased levels of burnout increase the occurrence of life events, whether life events increase the severity of burnout, or if both possibilities do occur. Alternatively, considering the effect of religion on burnout [64], low depersonalization may be related to the increased level of religiosity in our sample. Brazil is highly religious; 90% of our sample reported a religious affiliation, and religion has been reported to be a protective factor against burnout [51]. Because of this high level of religiosity, the generalization of our findings regarding religion to other cultures deserves further investigation. In addition, a sample from a teaching setting may also have some specificities. For example, the race distribution of our sample considering white, mixed and black was respectively, 261 (51.0%), 146 (28.5%) and 102 (19.9%), while a study reported that in Brazil that distribution was respectively, 37.6%, 44.5% and 12.9% [65]. This means that our sample was overrepresented by white and black nursing assistants. Working at a teaching hospital could also have contributed to the relatively reduced depersonalization scores of our sample compared to other Brazilian samples [66]. 

## 6. Conclusions

In aggregate, our data highlight the relevance of factors that affect personal vulnerability to burnout. Specifically, we found, in nursing assistants, that domains of life events in personal changes or difficulties were associated with high levels of emotional exhaustion. Life events in family situation and of death of friends or relatives were associated with high levels of depersonalization, independently of MDD and other potential confounders. Such data support the concept that, although related to workplace factors, burnout may also be influenced by personal vulnerabilities related to stressors outside the workplace. The relationship may be bi-directional for various life events; life events increase the risk of burnout, and burnout increases the risk of life events. Considering our findings, employers and managers aiming to reduce burnout in nursing professionals may consider the inclusion of programs focusing on the development of skills to improve coping strategies to deal with stressful life events and promotion of personal health activities, including spirituality, nutritional habits, and physical activity, particularly for nursing assistants. 

## Figures and Tables

**Table 1 ijerph-19-09342-t001:** Socio-demographic characteristics of nursing assistants.

Variables, Categorical	*N* (%)
Gender; *N* = 520	
Female	474 (91.2)
Male	46 (8.8)
Race ^a^; *N* = 512 ^b^	
White	261 (51.0)
Mixed	146 (28.5)
Black	102 (19.9)
Asian	3 (0.6)
Marital status; *N* = 520	
Married	260 (50.0)
Single	161 (31.0)
Separated	38 (7.3)
Divorced	28 (5.4)
Co-habitat	16 (3.1)
Widowed	17 (3.3)
N of children; *N* = 520	
0	164 (31.5)
1 or 2	259 (49.8)
≥3	97 (18.7)
Absenteeism ^c^; *N* = 518 ^b^	
0	408 (78.8)
≥1	110 (21.2)
Religion; *N* = 520	
Religion, yes	469 (90.2)
No religion	51 (9.8)
MDD; *N* = 519 ^b^	
MDD	138 (26.6)
No MDD	381 (73.4)
Variables, continuous	Mean (SD)
Age (years)	39.4 (9.7)
Weekly working hours	47.8 (20.6)
Years of work	8.5 (6.4)

MDD = major depressive disorder. ^a^ Race was self-reported based on the Brazilian census, including white, black, mixed, Asian and Indigenous people. ^b^ Number of subjects may differ because of missing data. ^c^ Missing days from work over the month prior to the interview.

**Table 2 ijerph-19-09342-t002:** Association between domains of life events and domains of burnout in nursing assistants, non-adjusted analysis.

Domains of Life Events	Domains of Burnout							
	Emotional Exhaustion				Depersonalization			
	Coefficient	95% Confident Interval		*p*	Coefficient	95% Confident Interval		*p*
Life events, total score	0.55	0.293	0.818	<0.001	0.140	0.0312	0.249	0.012
Troubles with law	1.050	0.051	2.048	0.039	0.233	−0.194	0.660	0.284
Work-related life events	0.812	0.184	1.44	0.011	−0.026	−0.301	0.248	0.849
Personal changes or difficulties	1.610	1.075	2.144	<0.001	0.243	0.015	0.470	0.036
Changes in environment	−0.007	−1.365	1.35	0.992	0.160	−0.376	0.697	0.558
Changes in familial situation	1.147	0.383	1.910	0.003	0.461	0.126	0.795	0.007
Death of relatives or friends	0.873	−0.840	2.586	0.317	0.911	0.165	1.657	0.017
Major depressive disorder	10.699	8.295	13.10	<0.001	2.293	1.230	3.356	<0.001
Gender	−0.946	−4.798	2.905	0.630	−0.902	−2.577	0.773	0.291
Age, years	−0.100	−0.211	0.010	0.076	−0.006	−0.056	0.042	0.783
Religion, yes	−4.165	−7.856	−4.750	0.027	−1.785	−3.379	−0.191	0.028
Kids, yes	−0.729	−3.141	1.682	0.553	0.016	−1.008	1.041	0.974
Physical activities, frequency per week	−0.518	−0.998	−0.0389	0.034	−0.090	−0.341	0.161	0.482
Personal income	0.009	−0.001	0.002	0.426	0.009	0.009	0.001	0.031
Time of working in the hospital, years	0.098	−0.069	0.265	0.249	0.092	0.0180	0.166	0.015

Domains of burnout were assessed with the Maslach Burnout Inventory-Human Services Survey. Domains of life events were assessed with the Social Readjustment Rating Scale.

**Table 3 ijerph-19-09342-t003:** Influence of life events of personal changes or difficulties on emotional exhaustion: Multiple linear regression analysis adjusted for confounders.

	Coefficient	Standard Error	*t*	*p*	95% Confidence Interval	
Personal changes or difficulties	1.523	0.366	4.16	<0.001	0.803	2.243
Major depressive disorder	9.171	1.218	7.53	<0.001	6.777	11.565
Religion, yes	−3.120	1.719	−1.81	0.070	−6.497	0.257
Physical activities, frequency per week	−0.431	0.268	−1.61	0.109	−0.958	0.095
Age	−0.097	0.054	−1.78	0.076	−0.205	0.010
Gender	−2.501	1.800	−1.39	0.165	−6.037	1.035
Work-related life events	−0.933	0.436	−2.14	0.033	−1.789	−0.076
Constant	26.478	3.157	8.39	0.000	20.275	32.681

An equation of multiple linear regression analysis was constructed with the domain of emotional exhaustion as the dependent variable and life events of personal changes or difficulties as the independent one. The model was adjusted for work-related life events and other potential confounders significantly related with emotional exhaustion in the bivariate analysis.

**Table 4 ijerph-19-09342-t004:** Influence of life events of changes in familial situation on depersonalization: Multiple linear regression analysis adjusted for confounders.

	Coefficient	Standard Error	*t*	*p*	95% Confidence Interval	
Changes in familial situation	0.458	0.183	2.50	0.013	0.098	0.818
Major depressive disorder	2.04	0.549	3.72	<0.001	0.961	3.119
Gender	−1.205	0.842	−1.43	0.153	−2.860	0.448
Age, years	−0.055	0.031	−1.75	0.081	−0.118	0.006
Personal income	0.009	0.009	1.79	0.074	−0.009	0.001
Religion, yes	−1.582	0.799	−1.98	0.048	−3.152	−0.012
Time of working in the hospital, years	0.089	0.050	1.78	0.076	−0.009	0.187
Work-related life events	−0.210	0.153	−1.38	0.169	−0.511	0.089
Constant	7.714	1.575	4.90	0.000	4.619	10.809

An equation of multiple linear regression analysis was constructed with the domain of depersonalization as the dependent variable and the life events of changes in familial situation as the independent one. The model was adjusted for work-related life events and other potential confounders significantly related to depersonalization in the bivariate analysis.

**Table 5 ijerph-19-09342-t005:** Influence of life events of death of relatives or friends on depersonalization: Multiple linear regression analysis adjusted for confounders.

	Coefficient	Standard Error	*t*	*p*	95% Confidence Interval	
Death of relatives or friends	0.744	0.374	1.99	0.048	0.008	1.481
Major depressive disorder	2.122	0.548	3.87	<0.001	1.045	3.199
Gender	−0.961	0.840	−1.14	0.253	−2.612	0.689
Age, years	−0.064	0.031	−2.03	0.043	−0.127	−0.002
Personal income	0.009	0.009	1.71	0.088	−0.009	0.001
Religion, yes	−1.544	0.800	−1.93	0.054	−3.116	0.028
Time of working in the hospital, years	0.097	0.050	1.95	0.051	−0.009	0.196
Work-related life events	−0.090	0.1428	−0.64	0.525	−0.371	0.189
Constant	8.119	1.568	5.18	0.000	5.037	11.200

An equation of multiple linear regression analysis was constructed with the domain of depersonalization as the dependent variable and the life events of death of relatives or friends as the independent one. The model was adjusted for work-related life events and other potential confounders significantly related to depersonalization in the bivariate analysis.

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
