# Peer review of "The Association of Life Events Outside the Workplace and Burnout: A Cross-Sectional Study on Nursing Assistants"

_ijerph, 2022, doi:10.3390/ijerph19159342_

Round 1
Reviewer 1 Report
The authors have investigated the impact of life events on burnout among a subpopulation of healthcare workers (assistant nurses). The study is relevant given the high burden that the COVID-19 pandemic inflicted in healthcare workers of all ranks and its results can be used in efforts to promote wellbeing in healthcare facilities.
The methods and the results of the study are sufficiently presented. Judging from the color differences in the manuscript, I understand that this manuscript must have already been revised based on previous peer - reviewers' comments. The current version seems suitable for publication.
I would advise the authors to proofread the manuscript once more for mistakes in spelling. For instance, in the abstract "whether live events outside the workplace", the term "life events" would be more appropriate.
Author Response
The authors have investigated the impact of life events on burnout among a subpopulation of healthcare workers (assistant nurses). The study is relevant given the high burden that the COVID-19 pandemic inflicted in healthcare workers of all ranks and its results can be used in efforts to promote wellbeing in healthcare facilities.
The methods and the results of the study are sufficiently presented. Judging from the color differences in the manuscript, I understand that this manuscript must have already been revised based on previous peer - reviewers' comments. The current version seems suitable for publication.
- I would advise the authors to proofread the manuscript once more for mistakes in spelling. For instance, in the abstract "whether live events outside the workplace", the term "life events" would be more appropriate.
Answer:
Dear reviewer, thank you for your advice, we performed a careful review of the entire manuscript looking for mistakes in spelling. We found other mistakes. Thank you.
Reviewer 2 Report
a) The authors made a research on the topic which is very popular nowadays. The authors describe the research object and its dimensions but deeper validations with the newest research could be made.
b) The authors use approved research scales for their research and the sample size seems appropriate. For analyses, the authors use linear regression analyses and I have doubts if this analysis is made correctly. I believe that regression analyses are made to construct a statistical model and some statistical variables must be discussed and provided accordingly. Please validate your used statistical analysis methods by applying references.
c) The authors deeply discuss their research results but it is not enough clear how their linear regression analysis results relate to their discussion.
Author Response
Reviewer 2
Dear reviewer, thank you for your comments and suggestions. They gave us an opportunity to improve our manuscript. Below are our answers.
- The authors made a research on the topic which is very popular nowadays. The authors describe the research object and its dimensions but deeper validations with the newest research could be made.
Answer:
We included the comment below in the discussion addressing the potential impact of the Covid-19 pandemic crisis.
4.5. Burnout, life events and the Covid-19 crisis
It should be considered that changes in contemporary external conditions includ-ing the Covid-19 pandemic crisis could interfere with the occurrence of life events and burnout. During the Covid-19 crisis, high burnout level was reported for depersonali-zation, emotional exhaustion, and personal accomplishment respectively in 47%, 46%, and 48% of 1,925 emergency medicine professionals including physicians, nurses, and paramedics [60]; in Brazil a great impact of Covid-19 in depersonalization and emo-tional exhaustion has been reported in a sample composed by physicians, nurses, nurse technologists and physiotherapists [61]. During the Covid-19, it has been found that the highest average scores and counts of stressful life events and highest work-family conflict levels were reported respectively by nurse practitioners and registered nurses compared to physicians and other professionals [41]. Also, students in the nursing field experienced more traumatic events during the Covid-19 pandemic crisis than before it [62]. The sense of coherence may act as a negative regulator between burnout and depression and it has been reported to be reduced by the influence of the Covid-19 pandemic [63]. Our study was developed before the Covid-19 pandemic crisis; thus, it is possible that during the Covid-19 crisis and also post Covid-19 crisis, other specific life events could show a relationship with burnout.
- The authors use approved research scales for their research and the sample size seems appropriate. For analyses, the authors use linear regression analyses and I have doubts if this analysis is made correctly. I believe that regression analyses are made to construct a statistical model and some statistical variables must be discussed and provided accordingly. Please validate your used statistical analysis methods by applying references.
Answer:
We included the text below in the Method section explaining that the multiple linear analysis was used to investigate the possible existence of confounders, explaining how the statistical equations were constructed and giving references.
We used simple linear regression analyses to investigate the associations without adjustments. Then, we constructed an equation of multiple linear regression to investigate the possible influence of confounders [40]. An equation was constructed for each domain of burnout significantly related to life events as the dependent variable. As the independent variable for each equation, we included the domains of life events that showed an association in the non-adjusted analysis. The equations were adjusted for work-related life events, gender, age, and variables potentially related to burnout showing significant association in the non-adjusted analysis. We used linear regression to investigate the relationship between life events and burnout as it has been used to determine the impact of stress on physical health and work-family conflict [41].
- c) The authors deeply discuss their research results but it is not enough clear how their linear regression analysis results relate to their discussion.
Answer
In the beginning of the discussion (below), we included the results that were from the multiple linear analysis. Thank you.
In this study with 521 nursing assistants, an increased number of life events outside the workplace was associated with increased severity of depersonalization and emotional exhaustion. The multiple linear analysis showed that an increased number of life events in the domain of personal changes or difficulties was associated with increased severity of emotional exhaustion, and an increased number of life events in the domains of familial situation and in the domain of death of relative or friends was associated with increased severity of depersonalization. The adjusted multiple linear analysis showed that these associations were independent of life events in the workplace, age, gender and other potential confounders.
Reviewer 3 Report
I read a very interesting study entitled "The association of life events outside the workplace and burnout: a cross-sectional study on nursing assistants". The topic of the manuscript is particularly important as it deals with the most vulnerable population of nurses, the assistants. The statistical treatment of the sample is particularly good, while on the other hand the sample selection could have been better. A major problem is that the study was conducted before the pandemic crisis, the pandemic tragically changed many of the variables the researchers are looking at, on the other hand the study data is very useful as a cutoff to control for the pandemic effect.
Here are my observations:
Title: I would ask the authors to consider whether it is useful to add to the title that the study was done before the pandemic.
Method: The sample was not selected using a randomized method, however what matters most is whether the sample is representative. If it is easy, I would suggest comparing the sample (in terms of gender, years of work, etc.) to the general population of Brazilian nurses.
Results: It would be useful to compare the burnout averages found in this study in nursing assistants with similar studies measuring burnout in nurses in Brazil.
Discussion : In the discussion paragraphs, you advance the view that the employee's internal characteristics (Lines 335-338) and life events outside the workplace (Lines 282-283), determine the presence of burnout. These views were popular before the pandemic crisis. Major shortages of materials, understaffing of nurses and tremendous pressure on hospitals, (work environment), during the pandemic have tragically increased burnout levels in health care staff, I would ask that the discussion be updated. (https://journals.plos.org/plosone/article?id=10.1371/journal.pone.0260410
https://pubmed.ncbi.nlm.nih.gov/35620812/
https://www.mdpi.com/2227-9032/10/1/134 )
Author Response
Dear reviewer, thank you for your comments and suggestions, the manuscript improved with their inclusion.
I read a very interesting study entitled "The association of life events outside the workplace and burnout: a cross-sectional study on nursing assistants". The topic of the manuscript is particularly important as it deals with the most vulnerable population of nurses, the assistants. The statistical treatment of the sample is particularly good, while on the other hand the sample selection could have been better. A major problem is that the study was conducted before the pandemic crisis, the pandemic tragically changed many of the variables the researchers are looking at, on the other hand the study data is very useful as a cutoff to control for the pandemic effect.
Here are my observations:
1-Title: I would ask the authors to consider whether it is useful to add to the title that the study was done before the pandemic.
Answer
Dear reviewer, thank you for your observation, we agree that the fact that the data is pre Covid should be better addressed in the manuscript. We opted to include a topic specifically addressing this point in the discussion.
4.5. Burnout, life events and the Covid-19 crisis
It should be considered that changes in contemporary external conditions including the Covid-19 pandemic crisis could interfere with the occurrence of life events and burnout. During the Covid-19 crisis, high burnout level was reported for depersonalization, emotional exhaustion, and personal accomplishment respectively in 47%, 46%, and 48% of 1,925 emergency medicine professionals including physicians, nurses, and paramedics [60]; in Brazil a great impact of Covid-19 in depersonalization and emotional exhaustion has been reported in a sample composed by physicians, nurses, nurse technologists and physiotherapists [61]. During the Covid-19, it has been found that the highest average scores and counts of stressful life events and highest work-family conflict levels were reported respectively by nurse practitioners and registered nurses compared to physicians and other professionals [41]. Also, students in the nursing field experienced more traumatic events during the Covid-19 pandemic crisis than before it [62]. The sense of coherence may act as a negative regulator between burnout and depression and it has been reported to be reduced by the influence of the Covid-19 pandemic [63]. Our study was developed before the Covid-19 pandemic crisis; thus, it is possible that during the Covid-19 crisis and also post Covid-19 crisis, other specific life events could show a relationship with burnout.
2-Method: The sample was not selected using a randomized method, however what matters most is whether the sample is representative. If it is easy, I would suggest comparing the sample (in terms of gender, years of work, etc.) to the general population of Brazilian nurses.
Answer
Thank you, we included the comment below about specificities of our sample including demographic and burnout aspects in the limitations section.
In addition, a sample from a teaching setting may also have some specificities. For ex-ample, the race distribution of our sample considering white, mixed and black was re-spectively 261 (51.0%), 146 (28.5%) and 102 (19.9%), while a study reported that in Bra-zil that distribution was respectively 37.6% 44.5% and 12.9% [65], which means that our sample was overrepresented by white and black nursing assistants. Working at a teaching hospital could also have contributed to the relatively reduced depersonaliza-tion scores of our sample compared to other Brazilian samples [66].
Results: It would be useful to compare the burnout averages found in this study in nursing assistants with similar studies measuring burnout in nurses in Brazil.
Answer:
Thank you, as we answered above, we included the comment below about specificities of our sample including demographic and burnout aspects in the limitations section
In addition, a sample from a teaching setting may also have some specificities. For example, the race distribution of our sample considering white, mixed and black was respectively 261 (51.0%), 146 (28.5%) and 102 (19.9%), while a study reported that in Brazil that distribution was respectively 37.6% 44.5% and 12.9% [65], which means that our sample was overrepresented by white and black nursing assistants. Working at a teaching hospital could also have contributed to the relatively reduced depersonalization scores of our sample compared to other Brazilian samples [66].
Discussion : In the discussion paragraphs, you advance the view that the employee's internal characteristics (Lines 335-338) and life events outside the workplace (Lines 282-283), determine the presence of burnout. These views were popular before the pandemic crisis. Major shortages of materials, understaffing of nurses and tremendous pressure on hospitals, (work environment), during the pandemic have tragically increased burnout levels in health care staff, I would ask that the discussion be updated. ttps://journals.plos.org/plosone/article?id=10.1371/journal.pone.0260410https://pubmed.ncbi.nlm.nih.gov/35620812/https://www.mdpi.com/2227-9032/10/1/134)
Answer:
We agree that the pandemic crisis should be addressed in the manuscript. We included the topic below adressing this topic in the discussion as we mentioned above.
4.5. Burnout, life events and the Covid-19 crisis
It should be considered that changes in contemporary external conditions including the Covid-19 pandemic crisis could interfere with the occurrence of life events and burnout. During the Covid-19 crisis, high burnout level was reported for depersonalization, emotional exhaustion, and personal accomplishment respectively in 47%, 46%, and 48% of 1,925 emergency medicine professionals including physicians, nurses, and paramedics [60]; in Brazil a great impact of Covid-19 in depersonalization and emotional exhaustion has been reported in a sample composed by physicians, nurses, nurse technologists and physiotherapists [61]. During the Covid-19, it has been found that the highest average scores and counts of stressful life events and highest work-family conflict levels were reported respectively by nurse practitioners and registered nurses compared to physicians and other professionals [41]. Also, students in the nursing field experienced more traumatic events during the Covid-19 pandemic crisis than before it [62]. The sense of coherence may act as a negative regulator between burnout and depression and it has been reported to be reduced by the influence of the Covid-19 pandemic [63]. Our study was developed before the Covid-19 pandemic crisis; thus, it is possible that during the Covid-19 crisis and also post Covid-19 crisis, other specific life events could show a relationship with burnout.
This manuscript is a resubmission of an earlier submission. The following is a list of the peer review reports and author responses from that submission.
Round 1
Reviewer 1 Report
This manuscript aimed to study aimed to “investigate the possible influence of life events outside the workplace on levels of burnout in a nursing assistant sample”. The researchers completed a cross-sectional study in a single Brazilian university hospital with 521 nursing assistants. The manuscript is well organized. Validated tools were used to measure the concept of interest. The results present interesting data and contributions to the field of knowledge, specifically among nursing assistants. Notwithstanding, the current version of the manuscript has some issues that should be improved before considering this work for publication.
Major observations
- The study period and data collection method (i.e., online survey, paper survey, or mix) were not mentioned in the manuscript. In addition, there are no details about how the survey was distributed to potential participants and if reminders were sent. Please provide these details.
- The methods section describes the model-building process used in the study. However, the results section does not mention anything regarding the backward logistic regression process. Please briefly describe the results of the building model strategy.
- Work-related events could affect the association between burnout and life events. It is strongly recommended to explore if work-related events are confounding any of the described associations. For example, a sensitivity analysis could explore the potential confounding effect of major work-related events in the studied associations.
- Please justify why in section 3.1 why you excluded in the data analysis life events related to work and those with evident positive valance. In addition, some excluded life events (e.g., imprisonment, major mortgage, or violation of the law) could also influence the burnout of individuals. Please revise and expand.
- The manuscript presents some of the study limitations in lines 320-327. However, this section does not refer to the inherent limitations of the cross-sectional design, the single-center nature of the study, or the response rate. In addition, this section should discuss the limitations of causal inference. Furthermore, the current version of the manuscript is not discussing the potential confounding effect of major work-related events on the associations explored in this study.
Minor observations
- As reported in line 98, the response rate appeared to be 35.3% (521\1477). If so, report this number as the study response rate at the beginning of the results section of the manuscript, as well as in the abstract.
- It is stated in line 50 “erroneous placement in the media.” I suggest revising the wording of this sentence. Do you mean “posts in the social media of erroneous placement and nursing errors”?
- The sentence in lines 53 and 54 is affecting the flow of ideas, specifically “Here we investigate the association 53 between life events outside the workplace and burnout in nursing assistants.”
- I suggest revising the part that says “during the period of this research” in line 55. For clarity, this statement could say “Nursing assistants in Brazil are involved in basic bedside care, as well as …”
- Line 168 mentions “lack of” religious belief as one of the controlling variables. I suggest revising the wording of this specific variable to a non-judgemental term, for example: “no religion or faith” or “no affiliation to a church or faith”.
- Define ICHC-FMUSP in line 169.
- Please limit the OR values and their corresponding 95%CIs to two decimal points. Make sure this adjustment is reflected across all the sections of the manuscript.
- It was reported in the footnote of Table 2 that “Analysis was adjusted for potential confounders including gender, years of work and major depressive disorder”. I recommend including in this table the adjusted ORs of these variables. Similarly, I recommend including in Table 3 the adjusted ORs of control variables included in the models.
Author Response
Reviewer 1
This manuscript aimed to study aimed to “investigate the possible influence of life events outside the workplace on levels of burnout in a nursing assistant sample”. The researchers completed a cross-sectional study in a single Brazilian university hospital with 521 nursing assistants. The manuscript is well organized. Validated tools were used to measure the concept of interest. The results present interesting data and contributions to the field of knowledge, specifically among nursing assistants. Notwithstanding, the current version of the manuscript has some issues that should be improved before considering this work for publication.
Dear Reviewer
Thank you for the careful revision of our manuscript. We addressed your suggestions and the manuscript improved significantly with them.
Major observations
- The study period and data collection method (i.e., online survey, paper survey, or mix) were not mentioned in the manuscript. In addition, there are no details about how the survey was distributed to potential participants and if reminders were sent. Please provide these details.
Answer
We added the following sentences in the section “Participants” to clarify this topic:
“They were assessed face-to-face by a psychiatrist or a psychologist researcher during the period June to November 2009, at the general medical units (i.e., general internal medicine, hematology, endocrinology, pulmonary, nephrology, immunology, rheumatology, gastroenterology, and geriatric units), surgical unit, emergency room and intensive care unit. The researchers went to every unit and asked for volunteers. Equivalent numbers of nursing assistants were recruited from the day shift (i.e., 7:00 a.m. to 7:00 p.m.) and the night shift (i.e., 7:00 p.m. to 7:00 a.m.). Nursing assistants represent 76.9% of the nursing workforce staff, and our sample comprised 35.3% (521\1477) of all nursing assistants when the study was conducted at a General Unit of a University Hospital in Sao Paulo, Brazil”
- The methods section describes the model-building process used in the study. However, the results section does not mention anything regarding the backward logistic regression process. Please briefly describe the results of the building model strategy.
Answer
We rewrote the statistical analysis section and clarified how we constructed the equations. Please note that now we first used the burnout subscales without splitting them in tertiles and did not use the stepwise logistic regression in response to a reviewer observation.
Below is the passage regarding the construction of the equations:
“Then, we constructed equations of multiple linear regression using each domain of burnout significantly related to life events as the dependent variable. As the independent variables for each equation, we included the domains of live events that showed an association in the non-adjusted analysis. The equations were adjusted for work-related life events, gender, age and variables potentially related to burnout showing significant association in the non-adjusted analysis.”
- Work-related events could affect the association between burnout and life events. It is strongly recommended to explore if work-related events are confounding any of the described associations. For example, a sensitivity analysis could explore the potential confounding effect of major work-related events in the studied associations.
Answer
We included the work-related events and adjusted the equations for them as described in item 2, above. The relationship between the life events and burnout was independent of the work-related life events as described in the results:
“The multiple linear regression analysis indicated that burnout was influenced by life events in the domain of personal changes and difficulties, changes in familial situation and death of relative or friends, independently of work-related life events and potential confounders.”
- Please justify why in section 3.1 why you excluded in the data analysis life events related to work and those with evident positive valance. In addition, some excluded life events (e.g., imprisonment, major mortgage, or violation of the law) could also influence the burnout of individuals. Please revise and expand.
Answer
We now included work-related life events as a potential confounder and also investigated the relationship between life events in the domain of legal troubles and burnout and made the respective changes in the methods, statistical analysis and results. We included the following description of legal troubles in the methods:
“ e) legal problems (e.g., imprisonment, major mortgage, foreclosure of mortgage or loan, trouble with in-laws, minor mortgage or loan, and minor violation of law).”
- The manuscript presents some of the study limitations in lines 320-327. However, this section does not refer to the inherent limitations of the cross-sectional design, the single-center nature of the study, or the response rate. In addition, this section should discuss the limitations of causal inference. Furthermore, the
current version of the manuscript is not discussing the potential confounding effect of major work-related events on the associations explored in this study.
Answer
Thank you for the observation. We included the following passage in the limitation`s section
“Our sample is comprised of nursing assistants from a single site, a teaching hospital and studies are needed to confirm the extension of our findings for nursing assistants in other settings. We asked for volunteers from each unit until completing the relative proportion of nursing assistants from that unit. Thus, we are not able to estimate the percentage of nursing assistants that may have been interested to participate after we completed our target of responders from each unit. The cross-sectional design of our study limits causal inference of the relationships reported, and longitudinal studies are necessary to investigate whether increased levels of burnout increase the occurrence of life events or life events increase the severity of burnout or both possibilities do occur.”
Minor observations
- As reported in line 98, the response rate appeared to be 35.3% (521\1477). If so, report this number as the study response rate at the beginning of the results section of the manuscript, as well as in the abstract.
Answer
That is not the response rate, the assessments were performed face-to-face in each unit. We asked for volunteers from each unit until completing the relative proportion of nursing assistants from that unit. Thus, we are not able to estimate the percentage of nursing assistants that may have been interested to participate after we completed our target of responders from each unit. We included that limitation in the paragraph talking about study limitations as described in the item 5 above.
- It is stated in line 50 “erroneous placement in the media.” I suggest revising the wording of this sentence. Do you mean “posts in the social media of erroneous placement and nursing errors”?
Answer
We changed the passage to make that clear:
“There is no specific study with nursing assistants in Brazil, but registered nurses (RNs) face problems such as lack of recognition of the scientific aspect of nursing, the relevance of nursing work not being adequately recognized by the media, and work overload.”
- The sentence in lines 53 and 54 is affecting the flow of ideas, specifically “Here we investigate the association between life events outside the workplace and burnout in nursing assistants.”
Answer
We agree, thank you. We eliminated that sentence.
- I suggest revising the part that says “during the period of this research” in line 55. For clarity, this statement could say “Nursing assistants in Brazil are involved in basic bedside care, as well as …”
Answer
Thank you, we changed
“Nursing assistants in Brazil, are involved in basic bedside care, as well as more complex activities including inserting bladder and gastric catheters and collecting material for laboratory tests under the supervision of a RN”
- Line 168 mentions “lack of” religious belief as one of the controlling variables. I suggest revising the wording of this specific variable to a non-judgemental term, for example: “no religion or faith” or “no affiliation to a church or faith”.
Answer
Thank you, we changed all the passages into no religion or faith.
- Define ICHC-FMUSP in line 169.
Answer
Thank you for the observation, we opted to say that it was a university hospital and omit the name of the institution.
- Please limit the OR values and their corresponding 95%CIs to two decimal points. Make sure this adjustment is reflected across all the sections of the manuscript.
Answer
We reduced the excessive numbers. Please note that now we analysed burnout without using cut-offs and performed linear analysis instead of logistic regression. Because of small numbers, we reduced into centesimal points.
- It was reported in the footnote of Table 2 that “Analysis was adjusted for potential confounders including gender, years of work and major depressive disorder”. I recommend including in this table the adjusted ORs of these variables. Similarly, I recommend including in Table 3 the adjusted ORs of control variables included in the models.
Answer
We included all the effect sizes in the tables. Please note that now we used linear regression as commented above.
Reviewer 2 Report
Burnout, by definition, is related to adverse chronic workplace stressors. Nursing assistants have a lower leadership status with unclear duties` boundaries, making them vulnerable to burnout. Little is known about the influence of stressors outside the workplace on burnout in these employees.
The authors investigated the association between burnout and life events outside the workplace in nursing assistants.
They proposed an observational, cross-sectional, single site study with 521 nursing assistants at a university hospital, we assessed burnout with the Maslach Inventory, and life events with the Social Readjustment Rating Scale.
They considered different variables and domains of life events involving death or disease of relatives or friends; changes in family context; changes in personal activities; and personal difficulties, as the independent variables.
Results were adjusted for potential confounders including gender, lack of religious belief or practice, years of work and depression.
They concluded that Life events outside the workplace may increase the levels of burnout in nursing assistants.
The article has strong merits.
With a pure academic spirit I suggest the following minor changes:
- Better explicit the aim “This study aimed to investigate the possible influence of life events outside the workplace on levels of burnout in a nursing assistant sample”. You have to better value your work which is much broader.
- You have two separated sections entitled participant, measures. I suggest to put them in the methods with the following paragraphs: (a) participants and procedures, (b) measurer. (c) statistical design.
- Perhaps a flow chart in the methods could aid the reader to follow better the design.
- I suggest to introduce the results with a few sentences describing the structure.
- Check the resolution of the tables and the text with reference to the standard of the MDPI.
- Insert a table with the acronysm
Author Response
Reviewer 2
Burnout, by definition, is related to adverse chronic workplace stressors. Nursing assistants have a lower leadership status with unclear duties` boundaries, making them vulnerable to burnout. Little is known about the influence of stressors outside the workplace on burnout in these employees. The authors investigated the association between burnout and life events outside the workplace in nursing assistants. They proposed an observational, cross-sectional, single-site study with 521 nursing assistants at a university hospital, we assessed burnout with the Maslach Inventory, and life events with the Social Readjustment Rating Scale. They considered different variables and domains of life events involving death of relatives or friends; changes in family context; changes in personal activities; and personal changes or difficulties, as the independent variables. Results were adjusted for potential confounders including gender, lack of religious belief or practice, years of work and depression. They concluded that Life events outside the workplace may increase the levels of burnout in nursing assistants. The article has strong merits.
Dear reviewer, thank you for your careful revision. We incorporated your comments and thy significantly enriched our manuscript. Below are our answers.
With a pure academic spirit I suggest the following minor changes:
1-Better explicit the aim “This study aimed to investigate the possible influence of life events outside the workplace on levels of burnout in a nursing assistant sample”. You have to better value your work which is much broader.
Answer
Thank you for the suggestion, we added that we looked if it was independent of confounders
“This study aimed to investigate the possible influence of life events outside the workplace on levels of burnout in a nursing assistant sample. Additionally, we investigated the presence of potential confounders in that relationship.”
2-You have two separated sections entitled participant, measures. I suggest to put them in the methods with the following paragraphs: (a) participants and procedures, (b) measurer. (c) statistical design.
Answer
Thank you, we followed your suggestion, we think that in this way it is easier for the reader to follow the rationale of the study.
3-Perhaps a flow chart in the methods could aid the reader to follow better the design.
Answer
We included the item study characteristics to help the reader follow the structure of the study.
“In this observational, cross-sectional, single-site study, we interviewed face-to-face 521 nursing assistants using the Primary Care Evaluation of Mental Disorders (PRIME-MD) and assessed burnout with the Maslach Burnout Inventory-Human Services Survey (MBI-HSS) and life events with the Social Readjustment Rating Scale (SRRS).”
4- I suggest to introduce the results with a few sentences describing the structure.
Answer
Thank you. Now we began the results with the following passage:
“In this observational, cross-sectional, single site study, the relationship between life events outside the workplace and burnout. We interviewed face-to-face 521 nursing assistants using the PRIME-MD and assessed burnout with the MBI-SS and life events with the SRRS.”
5-Check the resolution of the tables and the text with reference to the standard of the MDPI. Insert a table with the acronyms
Answer
Thank you we checked the resolution of the tables and text according to the MDPI and insert a table with acronyms after the references.
Reviewer 3 Report
The sample size is large; the study idea is interesting.
Abstract: “Nursing assistants have a lower leadership status with unclear duties` boundaries, making 25 them vulnerable to burnout.”, I’m not sure that this statement is necessary, as, in my opinion, it does not help the reader to get started to the topic. Rather, provide a brief statement as to why work-place unrelated stressors should adversely impact on (work-place-related) burnout. One could also claim that one person might be vulnerable in the area of work, but not in the area of coping with private life issues. Please report participants’ age and gender-ratio. Please specify the directions: “Life events related to personal activities (i.e., DECREASE? change in recreation TIME?, church activities?, HIGHER?LOWER? EXHAUSTING? social activities, POOR? sleeping habits, CRITICAL eating habits and revision’ of personal habits) were….”
Introduction; please add that as per January 1st 2022, with the introduction of the ICD-11, burnout is coded as QD85 https://www.who.int/news/item/28-05-2019-burn-out-an-occupational-phenomenon-international-classification-of-diseases.
The authors claim that one reason of increased risk of suffering from BU among nurses in Brazil might be due to the content and duration of the vocational training; please consider further reasons, such as personality traits, vulnerability, social constraints, along with premorbid psychiatric issues.
“…Little is known about the potential contribution of….”, ok, so, please report what is “little known” and deduce hypotheses. More specifically, based on the available literature, what kind of pattern of results do you expect and why? Please also specify, if and to what extent the present data expand upon the current literature.
Methods: “…sample have previously been reported previously [26,27]…”, please revise. Please clearly declare, if and to what extent data or parts of them have been already published.
References: Please translate non-English titles into English and put them [into brackets].
MBI-HSS; please spell out all abbreviations when introduced for the first time.
Statistics; for reasons of statistics and content, it is highly discouraged to categorizing continuous dimensions, above all with such a large sample size (Altman and Royston, 2006; MacCallum et al., 2002). Next, stepwise regression analyses are completely discouraged, as they lead to biased information. In this view, Your models should be based on theory; not guided by statistics themselves. Smith, G. (2018). Step away from stepwise. Journal of Big Data, 5(1), 32. Mundry, R., & Nunn, C. L. (2008). Stepwise model fitting and statistical inference: turning noise into signal pollution. The American Naturalist, 173(1), 119-123. Whittingham, M. J., Stephens, P. A., Bradbury, R. B., & Freckleton, R. P. (2006). Why do we still use stepwise modelling in ecology and behaviour? Journal of animal ecology, 75(5), 1182-1189.
Given this statistical issues, the authors should re-run their statistical analyses, modify the Result section, and adapt the Discussion secetion.
References
Altman, D.G., Royston, P., 2006. The cost of dichotomising continuous variables. BMJ (Clinical research ed.) 332(7549), 1080-1080.
MacCallum, R.C., Zhang, S., Preacher, K.J., Rucker, D.D., 2002. On the practice of dichotomization of quantitative variables. Psychol Methods 7(1), 19-40.
Author Response
Reviewer # 3
Dear reviewer, thank you for the careful revision of our manuscript. We addressed your comments and performed a new analysis according to the recommendation. We think that the manuscript improved in its reliability. Thank you. Below are the answers to your comments.
1-Abstract: “Nursing assistants have a lower leadership status with unclear duties` boundaries, making 25 them vulnerable to burnout.”, I’m not sure that this statement is necessary, as, in my opinion, it does not help the reader to get started to the topic. Rather, provide a brief statement as to why workplace unrelated stressors should adversely impact on (work-place-related)burnout. One could also claim that one person might be vulnerable in the area of work, but not in the area of coping with private life issues.
Answer
We agree, we changed the passage in the abstract:
“Burnout, by definition, is related to adverse chronic workplace stressors. Life events outside the workplace have been associated with increased risk of psychiatric morbidity. However, it is not known whether live events outside the workplace may increase the severity of burnout.”
and included a statement at the end of the introduction:
“Life events not specifically in workplace has been associated with increased risk of psychiatric diagnosis.”
2-Please report participants’ age and gender-ratio.
Answer
In the results, we included the passage:
“Of the 521 nursing assistants, most were female (91.2%; female/male =10.36) and the mean age was 39.4 (±9.7) years.”
3-Please specify the directions: “Life events related to personal activities (i.e., DECREASE? change in recreation TIME?, church activities?, HIGHER?LOWER? EXHAUSTING?social activities, POOR? sleeping habits, CRITICAL eating habitsand revision’ of personal habits) were….”
Answer
We changed that passage to clarify the direction of the relationship:
“We found an association of increase in the number of life events in the domain of personal changes or difficulties (e.g., the domain including personal injury or illness and/or sexual difficulties, change in recreation, church activities, social activities, sleeping habits, eating habits and revision of personal habits) and increase in the severity of emotional exhaustion. These results suggest that life events in church activities increase the severity of depersonalization. We also found that having a religion decrease the severity of depersonalization.”
4-Introduction; please add that as per January 1 st 2022, with the introduction of the ICD-11, burnout is coded as QD85https://www.who.int/news/item/28-05-2019-burn-out-an-occupational-phenomenon-international-classification-of-diseases.
Answer
Thank you for the actualization! In the introduction we included the passage:
“According to the International Code of Diseases-11 edition, (ICD 11) released in 2022, burnout has been coded among the “Factors influencing health status or contact with health services” (QF4Z) [1].”
5- The authors claim that one reason of increased risk of suffering from BU among nurses in Brazil might be due to the content and duration of the vocational training; please consider further reasons, such as personality traits, vulnerability, social constraints, along with premorbid psychiatric issues.
Answer
We agree that including the additional reasons improves the consistency of the statement. In the introduction we changed the passage:
“In a more inclusive view, burnout may result from the interaction of various factors including personal traits, emotion, work-related stress and previous psychiatric disorders (Ribeiro, Filho et al. 2014, Ramirez-Elvira, Romero-Bejar et al. 2021, Membrive-Jimenez, Gomez-Urquiza et al. 2022) Consequently it is theoretically possible that episodes of stress from other life domain could contribute to the development of burnout (Bianchi, Truchot et al. 2014).”
6-“…Little is known about the potential contribution of….”, ok, so, please report what is “little known” and deduce hypotheses. More specifically, based on the available literature, what kind of pattern of results do you expect and why? Please also specify, if and to what extent the present data expand upon the current literature.
Answer
We changed the passage and made it clear that there is no study specifically with nursing assistants, and now we state our expectations.
“However, no study has investigated the potential contribution of external life events for the severity of burnout specifically in nursing assistants. Considering the vulnerability of nursing assistants, we expect that life events outside workplace might significantly increase the levels of burnout in these professionals. If confirmed, knowing that life events outside the workplace increase levels of burnout in nursing assistants might allow the development of programs specifically devoted to assisting these workers dealing with life events and reducing the levels of burnout.”
7-Methods: “…sample have previously been reported previously[26,27]…”, please revise. Please clearly declare, if and to what extent data or parts of them have been already published.
Answer
We clarified the previous publications.
“Data from this sample regarding the influence of depression on validation of the MBI-HSS, but not about life events, have previously been reported elsewhere and in a master thesis (Trigo 2010, Trigo, de Freitas et al. 2018).”
8-References: Please translate non-English titles into English and put them [into brackets].
Answer
We translated the references into English and put them [into brackets]
9-MBI-HSS; please spell out all abbreviations when introduced for the first time.
Answer
We spelled out the abbreviations when they were introduced for the first time and included a table with the acronyms.
10- Statistics; for reasons of statistics and content, it is highly discouraged to categorizing continuous dimensions, above all with such a large sample size (Altman and Royston, 2006;MacCallum et al., 2002).
Answer
We agree, we re-run the analysis without using cut-offs and categorizing and made the respective changes in the statistical analysis, results, discussion and abstract. We think that this approach gives more consistency and reliability to our study. We included the analysis in tertiles in the supplementary material to give additional information about the magnitude of the potential clinical impact of life events on burnout (odds ratio).
11- Next, stepwise regression analyses are completely discouraged, as they lead to biased information. In this view, Your models should be based on theory; not guided by statistics themselves. Smith, G. (2018). Step away from stepwise. Journal of Big Data, 5(1), 32. Mundry, R., & Nunn, C.L. (2008). Stepwise model fitting and statistical inference: turning noise into signal pollution. The American Naturalist, 173(1), 119-123. Whittingham, M. J., Stephens, P. A., Bradbury, R. B., &Freckleton, R. P. (2006). Why do we still use stepwise modelling in ecology and behaviour? Journal of animal ecology, 75(5),1182-1189.
Given these statistical issues, the authors should re-run their statistical analyses, modify the Result section, and adapt the Discussion section.
Answer
We re-run the statistical analysis including only the variables of interest, without categorizing and included the respective changes in the methods, results, discussion, conclusions and abstract.
We also agree and re-run the analysis without using the stepwise regression. We include the life events driven by our hypothesis and adjusted the equations only for controlling for the potential confounders. We made the respective changes in the statistical analysis, results, discussion and abstract.